# Two gates mediate NMDA receptor activity and are under subunit-specific regulation

Johansen B. Amin[1,2,9], Miaomiao He [3,9], Ramesh Prasad [4], Xiaoling Leng [4], Huan-Xiang Zhou [4,5] & Lonnie P. Wollmuth [6,7,8] ✉

Kinetics of NMDA receptor (NMDAR) ion channel opening and closing contribute to their unique role in synaptic signaling. Agonist binding generates free energy to open a canonical gate at the M3 helix bundle crossing. Single channel activity is characterized by clusters, or periods of rapid opening and closing, that are separated by long silent periods. A conserved glycine in the outer most transmembrane helices, the M4 helices, regulates NMDAR function. Here we find that the GluN1 glycine mainly regulates single channel events within a cluster, whereas the GluN2 glycine mainly regulates entry and exit from clusters. Molecular dynamics simulations suggest that, whereas the GluN2 M4 (along with GluN2 pre-M1) regulates the gate at the M3 helix bundle crossing, the GluN1 glycine regulates a 'gate' at the M2 loop. Subsequent functional experiments support this interpretation. Thus, the distinct kinetics of NMDARs are mediated by two gates that are under subunit-specific regulation.

Ion channels exist in two general conformations: a closed, non-conducting state and an open, conducting state, where the channel forms a water-filled pore or permeation pathway that certain ions can cross and impact membrane excitability. 'Gates' are barriers that occlude the flux of ions in the non-conducting or closed state. They can arise from a variety of mechanisms[1], from a physical barrier where the pore is too narrow for ions to cross in its closed state to free energy barriers such as "dewetting" pores, where changes in hydrophobicity prevent charged ion entry[2–6].

AMPA (AMPAR) and NMDA (NMDAR) receptors are glutamate-gated ion channels or ionotropic glutamate receptors (iGluRs) that participate in numerous brain functions[7]. These receptors are highly modular, layered proteins (Fig. 1a) and are members of the pore-loop family of ion channels reflecting that their channel pore is formed in part by a non-membrane spanning pore loop, referred to as the M2 pore loop (Fig. 1b)[7–11]. In homology to other pore-loop channels such as K⁺, Na⁺, and Trp channels, the remainder of the permeation pathway is

formed by a transmembrane helix (i.e., M1) N-terminal to the pore loop, but mainly by an "inner" helix (i.e., M3) C-terminal to the pore loop (Fig. 1b)[7,12]. For other pore-loop channels, gates have been identified at the bundle crossing formed by the inner helices[13] as well as at the pore loop[4,5,14]. Interestingly, some K⁺ channels may have gates at the bundle crossing and the pore loop[4].

In iGluRs, the M3 helices form a canonical gate at the bundle crossing[8,15–18]. Binding of agonists, glutamate in the case of AMPARs and glutamate and glycine in the case of NMDARs, to the extracellular ligand-binding domain (LBD) (Fig. 1a) induces conformational changes that are transduced to this M3 bundle crossing, in part via the LILI motif (Fig. 1b)[18], leading to channel opening[19–26]. Some studies have suggested a 'gate' at the M2 loop[27,28] and mutations in the M2 loop can alter receptor gating[29–31]. Still, the significance of the M2 loop to receptor gating is unknown.

NMDARs are heterotetramers composed of two obligatory GluN1 and typically some combination of two GluN2(A-D) subunits.

[1]Graduate Program in Cellular and Molecular Pharmacology, Stony Brook University, Stony Brook, NY 11794-5230, USA. [2]Medical Scientist Training Program (MSTP), Stony Brook University, Stony Brook, NY 11794-5230, USA. [3]Graduate Program in Biochemistry and Structural Biology, Stony Brook University, Stony Brook, NY 11794-5230, USA. [4]Department of Chemistry, University of Illinois at Chicago, Chicago, IL 60607, USA. [5]Department of Physics, University of Illinois at Chicago, Chicago, IL 60607, USA. [6]Center for Nervous System Disorders, Stony Brook University, Stony Brook, NY 11794-5230, USA. [7]Department of Neurobiology & Behavior, Stony Brook University, Stony Brook, NY 11794-5230, USA. [8]Department of Biochemistry & Cell Biology, Stony Brook University, Stony Brook, NY 11794-5230, USA. [9]These authors contributed equally: Johansen B. Amin, Miaomiao He. ✉e-mail: lonnie.wollmuth@stonybrook.edu

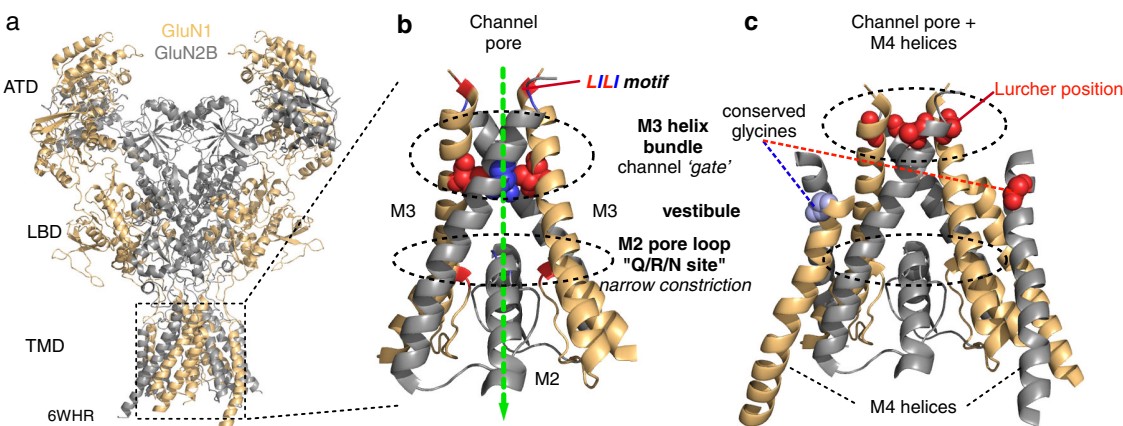

**Fig. 1 | Architecture of the NMDA receptors and their ion channel. a** NMDARs function as tetramers (2 GluN1/2 GluN2 subunits) with four domains in each subunit: extracellular amino-terminal (ATD) and ligand-binding (LBD) domains; transmembrane domain (TMD) forming the ion channel; and an intracellular C-terminal domain (not shown). GluN1/GluN2B, 6WHR[25]. **b** In homology to an inverted K⁺ channels, the ion channel core is the pore domain: transmembrane helices M1 ( = TM1 or S5) (not shown for clarity) and M3 (=TM2 or S6) and the M2

pore loop (= P loop). At the M3 helical apex is the 'helix bundle' that forms an external 'gate' that is mainly lined by residues in the highly conserved SYTANLAAF and regulated by the LILI motif. At the M2 loop apex is the N site (NMDARs) and the Q/R site (AMPARs). **c** The eukaryotic specific M4 transmembrane helix surrounds the pore domain (M1-M3) with the M4 of one subunit associating with the M1 and M3 of an adjacent subunit. Near the top of the M4 helices are glycines that are conserved across all mammalian iGluR subunits[37].

Surrounding the pore domain (M1-M3) (Fig. 1b) are eukaryotic specific M4 helices (Fig. 1c). The GluN1 and GluN2 M4 helices regulate the core gating machinery[32–36]. Critical in the M4 helices are highly conserved glycines positioned near the top of the M4s[37]. A variety of disease-associated missense mutations have been identified at or near these conserved glycines[38–40], and even the most conservative amino-acid substitution, glycine-to-alanine (G-to-A), dramatically alters receptor gating. Notably, the G-to-A substitution in the GluN1 but not GluN2 alters channel gating at least in part via constriction of the M2 pore loop which modifies Ca²⁺ permeation[37]. Additionally, studies of nearby missense mutations in the GluN2 M4 suggest that the M4 may act through the pore lining GluN1 M3 helices[38].

Here, we investigate the roles of the GluN1 and GluN2A M4 helices in regulating NMDAR function by focusing on the conserved glycines. The G-to-A substitution in either the GluN1 or GluN2A subunits dramatically curtailed channel opening[37]. However, we find that they have distinct effects on the overall pattern of single channel activity. The GluN2A G-to-A, which our molecular dynamics (MD) simulations indicate primarily alters the M3 gate, affects entry and exit from prolonged periods of activity known as "clusters". On the other hand, the GluN1 G-to-A, which our MD simulations show impedes ion flux through the M2 constriction, affects single channel opening within a cluster. These data suggest that the M3 gate mediates the longer opening that reflect entry into clusters, while the M2 gate mediates, at least in part, the faster gating activity within clusters. Reactivity of MTS reagents within the vestibule separating the M3 gate and the M2 loop further confirms the presence of a gate at the M2 constriction. Thus, our results support a two-gate model of NMDAR activity with these gates regulated by specific subunits and controlling different aspects of NMDAR activity.

## Results

A G-to-A substitution (or G-A for short) at a conserved glycine in the GluN1a but not GluN2 M4 helices affects Ca²⁺ permeation by altering the M2 loop constriction[37], suggesting that the M2 loop may function as a 'gate'. To address this notion, we characterized more extensively the single channel properties of the G-to-A substitutions at the conserved glycines in GluN1 and GluN2A.

### M4 helices regulate distinct components of NMDAR gating
We recorded in the on-cell configuration single channel activity of wild-type GluN1a/GluN2A, GluN1a(G815A)/GluN2A, and GluN1a/

GluN2A(G819A) for long durations, typically 20 min or longer (Fig. 2a and Supplementary Fig. 1a). For these experiments, we used saturating concentrations of co-agonists and recorded in the absence of divalents to enhance resolution (see "Methods" section). Both G-to-A substitutions dramatically reduce the equilibrium open probability (eq. $P_{open}$) to the same extent (Fig. 2b & Supplementary Table 1). The eq. $P_{open}$ is the fraction of time channels are open during the entire recording including during long-lived closed states. However, these constructs have differential effects on mean closed (MCT) and open (MOT) times with GluN1(G815A) (GluN1 G-A) notably reducing MOT (Supplementary Fig. 1b). To understand this difference, we characterized the effect of the G-As on the pattern of the single channel activity including clusters and interclusters (Fig. 2c–d and Supplementary Fig. 1c–e and Table 2).

Wild-type GluN1a/GluN2A shows a characteristic pattern of single channel activity: periods of high activity referred to as clusters (bursts, superclusters) (gray bars), during which the channel rapidly transitions between open and closed states, interspersed by periods of inactivity, which we refer to as interclusters (Fig. 2a, upper trace)[41,42]. Under our conditions, wild-type GluN1a/GluN2A showed a cluster duration around 12.3 sec (12.3 ± 1.0 s, n = 16; mean ± SEM, n = number of on-cell patches) and an intercluster duration around 2.3 s (2.3 ± 0.3 s; Supplementary Table 2).

While the G-to-A substitutions in GluN1a and GluN2A have the same effects on eq. $P_{open}$, they have different effects on cluster $P_{open}$ (Fig. 2c), which is significantly more reduced in GluN1a G-A than in GluN2A G-A with this effect related to a severely reduced MOT in GluN1 G-A (Supplementary Fig. 1c). Further, the G-As alter the cluster and intercluster pattern but do so in distinct ways with GluN2A G-A significantly reducing cluster length and extending intercluster length (Supplementary Fig. 1d, e). To contrast these differences, we characterized the probability of being in a cluster during the entire recording period ($P_{cluster}$) (Fig. 2d), which showed no difference between wild-type and GluN1 G-A but with GluN2A G-A being significantly reduced relative to both. Hence, the GluN2A M4 but not the GluN1a M4 regulates the cluster probability.

NMDAR single channel activity is characterized by five closed states[42–45], which we found to hold for wild-type as well as both G-to-A constructs (Supplementary Figure 2 & Table 3). Cluster analysis depends on the definition of a $T_{crit}$, which is a cut-off between the third ($C_3$) and fourth ($C_4$) closed kinetic states that minimizes false events[46] (see "Methods" section). To analyze the effects of the G-to-A constructs

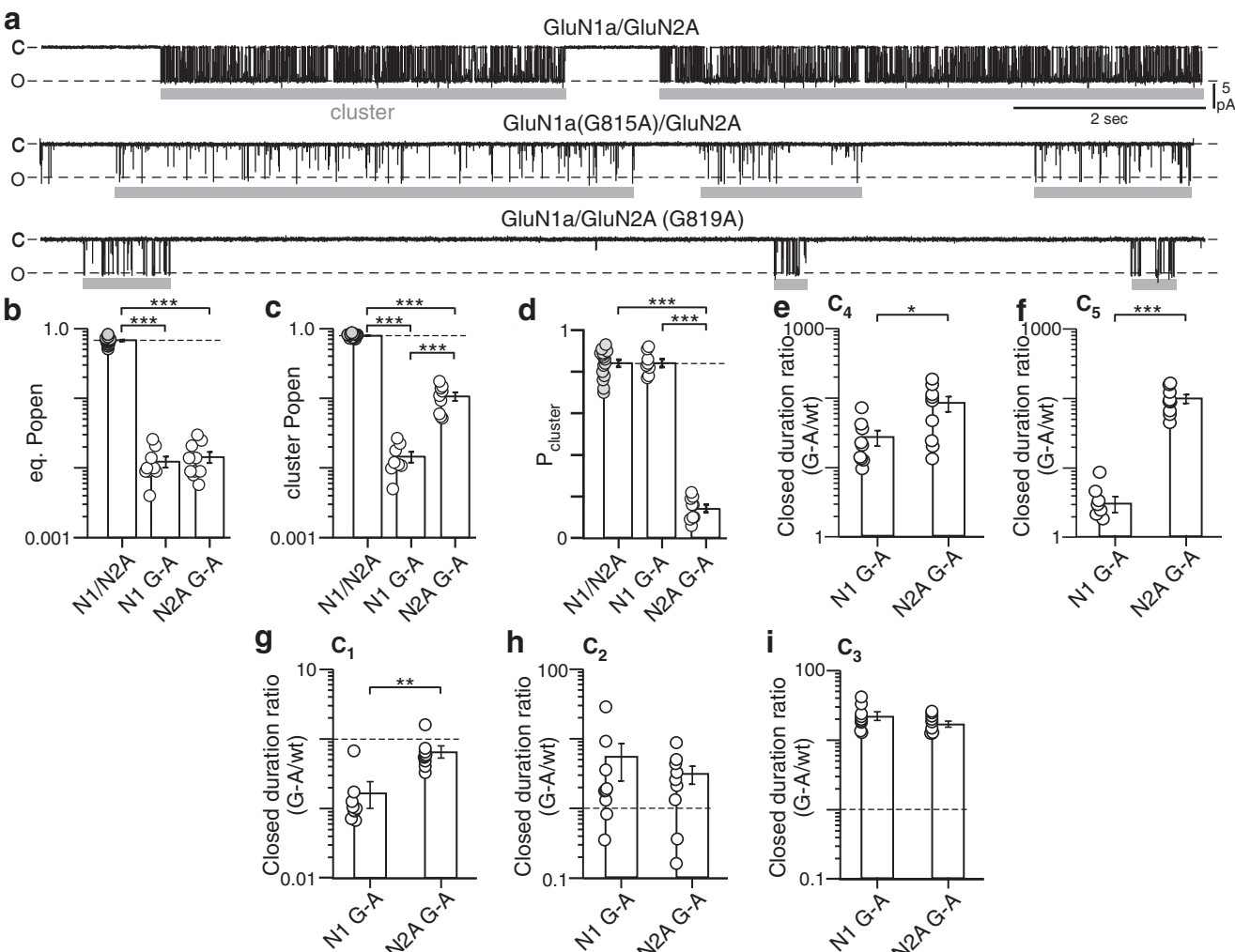

**Fig. 2 | GluN1 or GluN2A M4 helices have distinct effects on the pattern of NMDAR gating. a** Single channel traces of on-cell patches at −100 mV for wild-type (upper trace) or constructs containing alanine (A) substituted at the conserved glycine (G) either in GluN1a(G815A) (middle trace) or GluN2A(G819A) (lower trace). (C)losed and (O)pen states are indicated. Gray bars indicate 'clusters', which were defined based on $T_{crit}$. **b**–**d** Bar graphs (mean ± SEM with circles indicating individual values) of single channel equilibrium open probability (eq. $P_{open}$) (**b**), cluster $P_{open}$ (**c**), and cluster probability ($P_{cluster}$ = mean cluster duration/(mean cluster duration + mean intercluster duration) (**d**). **e**–**i** Ratio of mean duration (mean ± SEM) between G-A constructs and wild-type for the two slowest closed

components, $C_4$ (**e**) and $C_5$ (**f**) and for the three fastest components, $C_1$ (**g**), $C_2$ (**h**), and $C_3$ (**i**) (Supplementary Fig. 2). The dashed line is 1, which represents no effect relative to wild-type.\*\*$p < 0.01$, \*\*\*$p < 0.001$, one-way ANOVA with post-hoc Tukey's test (**b**–**d**) or two-tailed Student's $t$ test, unpaired (**e**–**i**). **b** ANOVA ($p = 8.2E-27$): wt vs N1 G-A, $p = 5.9E-15$; wt vs N2A G-A, $p = 5.8E-15$; N1 G-A vs N2A G-A, $p = 0.99$. **c** *ANOVA* ($p = 2.9E-29$): wt vs N1 G-A, $p = 4E-15$; wt vs N2A G-A, $p = 4E-15$; N1 G-A vs N2A G-A, $p = 0.0004$. **d** ANOVA ($p = 1.8E-22$): wt vs N1 G-A, $p = 0.97$; wt vs N2A G-A, $p = 1.6E-17$; N1 G-A vs N2A G-A, $p = 8.6E-14$. **e**–**i** Two-tailed Student's $t$ test, unpaired: $p = 0.028$ (**e**), 0.00013 (**f**), 0.0046 (**g**), 0.47 (**h**), and 0.24 (**i**). See Supplementary Tables 2 and 3 for ns and additional parameters.

on the gating pattern independent of defining $T_{crit}$, we compared the mean duration of the two longest closed states, $C_4$ and $C_5$ (Supplementary Fig. 1f, g). Consistent with the differential roles of the M4 helices in regulating the gating (cluster) pattern of NMDARs, the mean $C_4$ and $C_5$ durations for GluN2A G-A were consistently longer relative to both wild-type and GluN1 G-A. This difference is further revealed when the mean closed durations of the G-to-A substitutions are compared as ratios relative to wild-type, either for $C_4$ (Fig. 2e) or $C_5$ (Fig. 2f). In contrast, the shortest-lived closed state, $C_1$, was predominantly altered by GluN1 G-to-A (Fig. 2g and Supplementary Fig. 1h), whereas C2 (Fig. 2h) and C3 (Fig. 2i) were equally affected.

These results suggest that the M4 helices in GluN1 and GluN2A subunits contribute differently to the kinetics of NMDAR gating. The GluN2A M4s predominantly regulate a 'cluster' gate, dictating entry into and exit from long-lived closed states (Fig. 2d). In contrast, the GluN1 M4s predominantly regulate channel activity 'within' a cluster, impacting mainly the duration of open time (Supplementary Fig. 1b)

and hence cluster $P_{open}$ (Fig. 2c) as well as compressing the shortest-lived closed state (Fig. 2g).

## MD simulations suggest that the M2 pore loop acts as a 'gate'

The GluN1 M4 but not the GluN2 M4 regulates the pore radius of the M2 pore loop and consequently the magnitude of $Ca^{2+}$ permeation[37]. Given this association of the GluN1 M4 helix with the M2 pore loop, we carried out MD simulations on an 'open' state homology model for GluN1/GluN2B[37]. In part due to the uncertainty of the wild-type open state structure, we are mainly interested in outcomes for the G-to-A substitutions relative to our wild-type and refer to this approach as comparative MD simulations. Although the MD simulations were run on GluN1/GluN2B constructs, we note that GluN2B G-A and GluN2A G-A have the same relative effect on single channel activity[37]. With the LBD-TMD linkers 'locked' in a presumed open conformation, we ran 15 replicate simulations for 500 nanoseconds each and repeated the simulations after introducing G-A in either GluN1 or

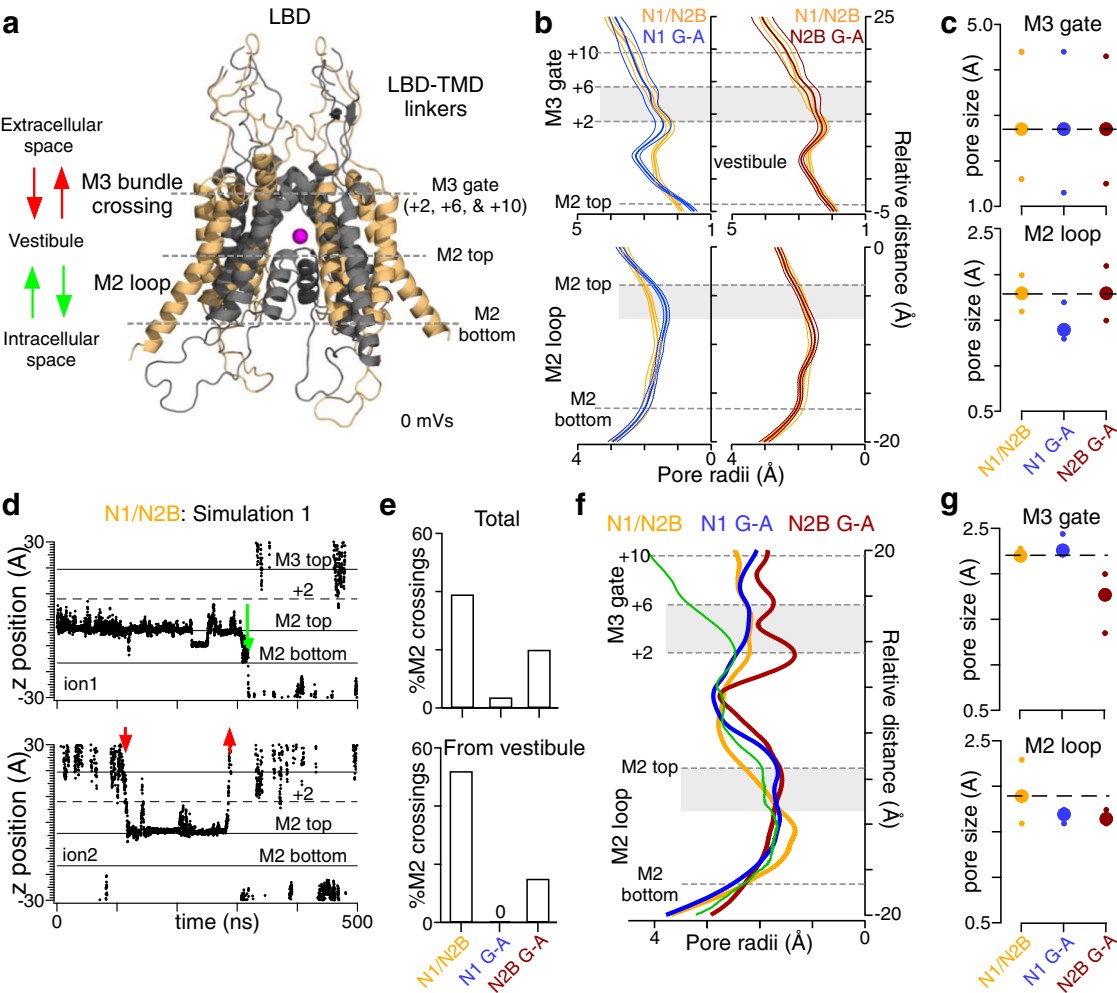

**Fig. 3 | Comparative molecular dynamic (MD) simulations demonstrate subunit specific regulation of the pore domain and that the M2 loop may act as a 'gate'. a** Structure of TMD used for simulations (Supplementary Fig. 3). Magenta sphere is a Na⁺ ion in the vestibule external to M2 loop. For simulations in **b–e**, LBD-TMD linkers are restrained to 'open' conformation. For **f, g** constraints are released. **b** Pore radius (mean ± SD, thick line ± thin lines) along the channel axis through M3 bundle crossing (upper) or M2 loop (lower) for wild type, and GluN1 G-A (left) or wild type and GluN2B G-A (right). Pore radii are average over the entire 500 ns. Gray shade indicates region used for calculating average (**c**). **c** Pore radius (average ± minimum & maximum) around M3 gate ($z = 8.8–14.1$ Å) (upper) and M2 loop ($z = −3.9$ to $−7$ Å) (lower). **d** Example ion trajectories for wild type (same simulation in Supplementary Fig. 4). Dots show location of a Na⁺ ion. Red and green arrows

indicate complete passages across M3 and M2 loop, respectively (inset panel **a**). **e** %M2 crossings (number of M2 crossings/total number) either for total number (upper) or for crossings starting in vestibule (lower). Values are from 15 replicate simulations. Numbers of M2 and M3 crossings: (upper) N1/N2B, 22, 34; N1 G-A, 3, 81; and N2B G-A, 7, 28; (lower) N1/N2B, 12, 11; N1 G-A, 0, 32; and N2B G-A, 2, 12. **f** Pore radii along the pore axis for wild-type N1/N2B, GluN1 G-A, and GluN2B G-A. At time '0' constraints on the LBD-TMD linkers that were keeping the TMD open were released. Green curve is the average pore diameter for the 3 constructs prior to constraint release. The other curves are pore diameters after 3000 ns of simulations without linker constraints. Gray shade indicates region used for calculating average (**f**). **g** Average pore radius (average ± minimum & maximum) around M3 (+8.8 to 14.1 Å) (upper panel) and M2 loop (−3.9 to −7 Å) (lower panel).

GluN2B (Fig. 3a–e and Supplementary Figs. 3 and 4). Achieving the open conformation for the LBD-TMD linkers requires significant rearrangements of the extracellular domain[47,48], which we did not explicitly model here. The ions in the simulations were monovalent (with Na⁺ as permeant ions), matching conditions in our single channel recordings.

We characterized the effect of the G-to-A substitutions on pore radius, focusing on the M3 helix bundle crossing, encompassing positions +2, +6, and +10, and the M2 pore loop (Fig. 1b and Supplementary Fig. 3a, b). The G-to-A substitutions in GluN1 or GluN2 did not affect pore radius at the M3 crossing (Fig. 3b, c, upper panels), reflecting that throughout the simulations the LBD-TMD linkers are locked in an open conformation. In contrast and consistent with previous simulations[37], the G-to-A substitution in GluN1, but not in GluN2, reduced the pore dimensions at the M2 loop (Fig. 3b, c, lower panels).

To address how these changes in pore radius might affect ion flow, we tracked the movement of Na⁺ ions across the pore during the simulations (Fig. 3d and Supplementary Fig. 4). We defined three compartments along the permeation pathway (Fig. 3a, inset): extracellular to the +10 position in the M3 helix; the vestibule between the M3 helix bundle crossing and the M2 pore loop; and intracellular, below the M2 loop. We classified 'permeation events' as full transitions either through the M3 bundle crossing (red arrows) or through the M2 loop (green arrows) (Fig. 3a, inset). At the start of each simulation, a pool of ions was available for permeation in the extracellular (+30 Å) or intracellular (−30 Å) space or in the vestibule. Ions once inside the vestibule could leave by traversing the M2 loop (downward green arrows; Fig. 3d) but more frequently by traversing the M3 bundle crossing (upward red arrows). Ions could also traverse from the extracellular (downward red arrows) or intracellular (upward green arrows) space to the vestibule (Fig. 3d and Supplementary Fig. 4).

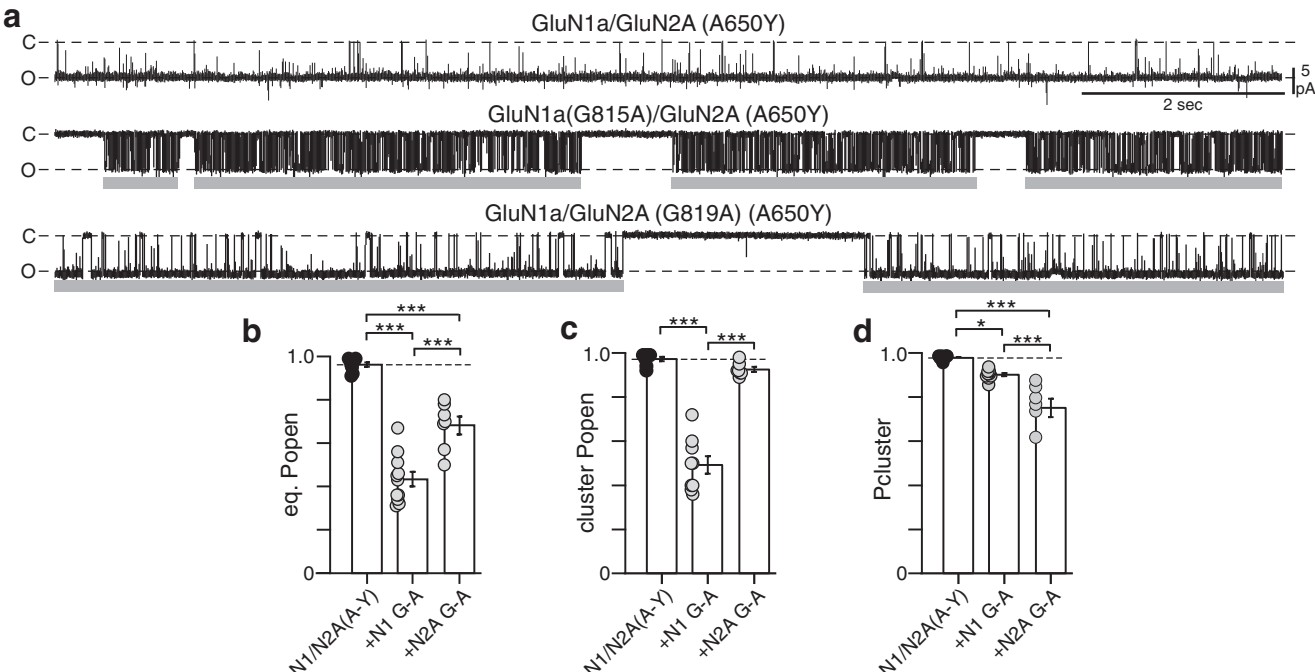

**Fig. 4 | Prying open the M3 gate. a** On-cell recordings of constructs containing a tyrosine (Y) substituted at A650 in GluN2A, either alone or with G-to-As at the conserved glycines. Records displayed and analyzed as in Fig. 2. **b–d** Bar graphs (mean ± SEM) for single channel equilibrium $P_{open}$ (**b**), cluster $P_{open}$ (**c**), and cluster probability ($P_{cluster}$) (**d**). *$p < 0.05$, ***$p < 0.001$, one-way ANOVA with post-hoc Tukey's test. **b** *ANOVA* ($p = 2.5E-11$): A-Y vs N1 G-A, $p = 1.3E-11$; A-Y vs N2A G-A, $p = 1.1E-05$; N1 G-A vs N2A G-A, $p = 3.2e-05$. **c** *ANOVA* ($p = 2.1E-11$): A-Y vs N1 G-A, $p = 7E-11$; A-Y vs N2A G-A, $p = 0.45$; N1 G-A vs N2A G-A, $p = 10E-10$. **d** *ANOVA* ($p = 5.4E-07$): A-Y vs N1 G-A, $p = 0.024$; A-Y vs N2A G-A, $p = 3.7E-07$; N1 G-A vs N2A G-A, $p = 10E-05$. See Supplementary Tables 5 and 6 for ns and additional parameters.

We characterized M2 crossings as a fraction of the total number of crossings (Fig. 3e, upper panel). Notably, M2 crossings represented a significant fraction of the total crossings for both wild-type, at 39% of total crossings, and GluN2A G-A, at 20% of total crossings. In contrast, M2 crossings were rare in GluN1 G-A, at only 3.7% of total crossings.

A potential complication is that, while random, more ions entered the vestibule from the extracellular space than from the intracellular space (Supplementary Fig. 4), thereby raising the number of M3 crossings. To address this concern, we analyzed those events where the ion started in the vestibule, with the assumption that all other things being equal there is a 50% chance of going either outward across the M3 gate or inward across the M2 loop (Fig. 3e, lower panel). For wild-type, 52% of total crossings were via the M2 loop (12 out of 23 total crossings), whereas for the GluN2A G-A 14% were via the M2 (2 out of 14 total crossing). In contrast, despite having a highest number of starting ions in the vestibule, 32 total, none of these ions crossed the M2 loop in the GluN1 G-A mutant, with all crossings at the M3 gate (Fig. 3e, lower panel).

For the simulations reported in Fig. 3, no transmembrane potential was present. We carried out additional simulations applying a −300 mV transmembrane potential (16 replicates of 250-ns duration for each construct). As expected, the transmembrane potential increases the number of crossings (M3 or M2) for all constructs. Importantly, the percentage of M2 crossings relative to the total number of crossings in GluN1 G-A is still strongly attenuated (15.7%, 16 out of 102 total crossings) compared to either wild-type (46.6%, 41 out of 88) or GluN2 G-A (36.4%, 24 out of 66).

In summary, these MD simulations are consistent with the idea that the M2 loop can act as a 'gate', controlling the flux of ions across the channel and, moreover, that the GluN1 M4 regulates this M2 gate. A further notable outcome is that the M2 gate can close while the M3 gate is in a fully open conformation (Fig. 3c). Hence, the M2 gate can transition between the presumed open and closed conformation while the M3 gate is open.

## The GluN2 M4 helix regulates the M3 gate

For the MD simulations in Fig. 3b–e, we restrained the LBD-TMD linkers in a presumed open conformation, which prevented any major change in the M3 bundle crossing (Fig. 3b, c, upper panels). To assess how the M4 helices might impact the M3 gate, we released the restraints on the LBD-TMD linkers and carried out MD simulations for 3000 ns and averaged the pore radii over the last 1500 ns for each construct (Fig. 3f, g).

The starting pore radii for all three constructs before restraint release were comparable and we use their average as reference (thin green line, Fig. 3f). After the LBD-TMD linker restraints were removed, the most dramatic change was a collapse of the M3 gate (+2 to +10) for GluN2B G-A. While this effect was most pronounced around the +2 position, we averaged the pore radii from +2 to +6 (Fig. 3g, upper panel): the GluN2B G-A showed a strong reduction in pore radius while there was no difference between wild-type and GluN1 G-A. In contrast, averaging across the M2 pore loop did not reveal a significant difference between the constructs (Fig. 3g, lower panel), although wild-type did show some collapse around −12 Å. Hence, the GluN2 M4 impacts the M3 gate whereas GluN1 M4 does not.

## Locking the M3 gate 'open' suggests two distinct gates

Studying a potential 'gate' at the M2 loop is challenging since it will be intertwined with the external M3 gate, which if closed would preclude access to this internal gate. To circumvent this problem, we took advantage of a tyrosine (Y) substitution at an alanine adjacent to the Lurcher position in GluN2A (A650Y) that dramatically enhances receptor gating[49]. Since this position is part of the M3 gate[7,8], we assume that A650Y locks the M3 gate in an 'open' conformation, comparable to our MD simulations where the LBD-TMD linkers locked the channel open (Fig. 3b–e). Consistent with this idea, GluN1a/GluN2A(A650Y) channels showed extremely high eq. $P_{open}$, about 0.96 (0.96 ± 0.01, $n = 9$) compared to 0.68 for wild-type and long MOTs, about 40 ms (40 ± 5 ms) compared to 6.4 ms for wild-type

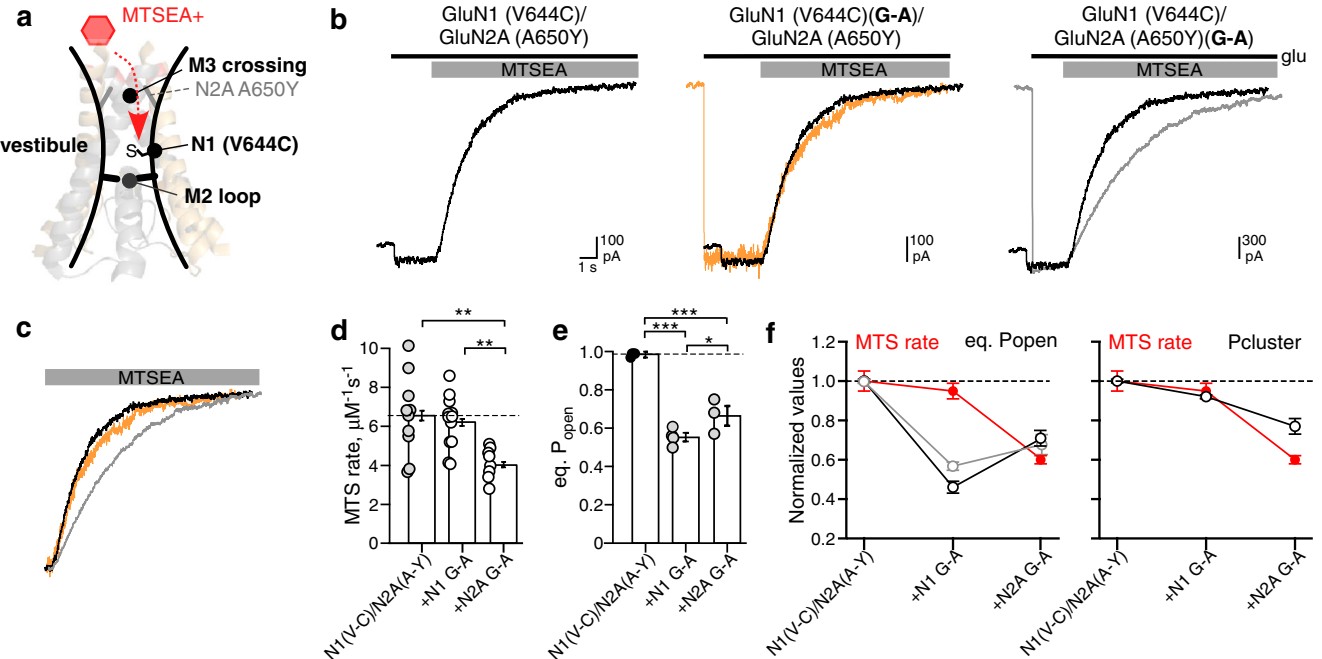

**Fig. 5 | Rates of MTS reactivity indicate that the M2 loop forms a gate. a** Cartoon illustrating the position of GluN1 V644 in the vestibule between the M3 gate and the M2 loop. **b** Whole-cell recordings of constructs containing a cysteine substituted in the vestibule, GluN1a(V644C), either alone (*left panel*) or with GluN1a(G815A) (*center panel*) or GluN2A(G819A) (*right panel*). Currents were activated by glutamate (black line) and then cells were exposed to MTSEA (75 μM). **c** Aligned MTSEA-induced current decay for the three constructs. **d, e** Bar graphs (mean ± SEM) for MTS reaction rates (**d**) or equilibrium $P_{open}$ (**e**) for cysteine containing constructs. See Supplementary Table 7 for ns and additional parameters. **f** Values normalized to control (mean ± SEM) for MTS reaction rates (red) and equilibrium $P_{open}$ (left

panel) or $P_{cluster}$ (right panel). Single channel properties were from constructs either with (gray) or without (black) cysteines. For ns and statistics: MTS reactivity, **d**; eq. $P_{open}$ for constructs with cysteines, **e**; eq. $P_{open}$ for constructs without cysteines, Fig. 4b; and $P_{cluster}$ for constructs without cysteines, Fig. 4d. Note because of the low number of ns and the overlap of eq. $P_{open}$ (**f**, left panel), we did not do cluster analysis for constructs with cysteines. *$p < 0.05$, **$p < 0.01$, ***$p < 0.001$, one-way ANOVA with post-hoc Tukey's test. **d** ANOVA ($p = 0.0019$): V-C/A-Y vs N1 G-A, $p = 0.86$; V-C/A-Y vs N2A G-A, $p = 0.0029$; N1 G-A vs N2A G-A, $p = 0.0071$. **e** ANOVA ($p = 1.3E-06$): V-C/A-Y vs N1 G-A, $p = 1.3E-06$; V-C/A-Y vs N2A G-A, $p = 3.2E-05$; N1 G-A vs N2A G-A, $p = 0.041$.

(Fig. 4a, upper panel, 4b and Supplementary Fig. 5 and Table 5). GluN1a/GluN2A(A650Y) did visit the closed state, but these transitions were typically extremely brief $C_1$ and $C_2$ closures (Supplementary Figs. 6 and 7 and Table 6).

To assay the impact of the GluN1a and GluN2A M4 helices on this A650Y-induced gating pattern, we introduced the G-As in the GluN2A(A650Y) background (Fig. 4a–d; Supplementary Figs. 5–7 and Tables 5–8). The effects of these substitutions in the A650Y background were largely comparable to those in the wild-type background (Fig. 2a–i), including significant reduction in eq. $P_{open}$ (Fig. 4b and Supplementary Fig. 5b and Table 5), greater reduction in cluster $P_{open}$ by GluN1a G-A (Fig. 4c), and greater reduction in $P_{cluster}$ by GluN2A G-A (Fig. 4d, Supplementary Figs. 5c–e and 6, and Table 6). These comparable outcomes support the idea of the distinct roles of the GluN1 and GluN2 M4s in receptor function.

**MTS modification rates support two gates**

A salient effect of the G-to-A substitutions in the A650Y background was to induce a significant reduction in eq. $P_{open}$ (Fig. 4b), with this effect much greater for GluN1 G-A. We hypothesize that these decreased open probabilities reflect increased closures of the M3 gate (by GluN2A G-A) and the M2 gate (by GluN1 G-A). To test this hypothesis, we quantified the rate of reactivity of the MTS reagent aminoethyl-methanethiosulfonate (MTSEA) with a cysteine substituted in the external vestibule, GluN1(V644C), located between the M3 gate and the M2 loop (Fig. 5a), in the GluN2A(A650Y) background. Given this positioning of the introduced cysteine, closures of the external or upstream M3 gate would slow the rate of reactivity whereas closures of the downstream M2 gate would have no effect on the reactivity rate.

Consistent with our hypothesis, the rate of MTS reactivity with GluN1 G-A was indistinguishable from the control A650Y background (Fig. 5b–d) despite GluN1 G-A showing a stronger decrease in eq. $P_{open}$ for constructs either lacking cysteines (Fig. 4b) or with cysteines (Fig. 5e and Supplementary Table 9). In contrast, GluN2A G-A showed a significantly reduced MTS reactivity rate (Fig. 5b–d), again consistent with the idea that it is affecting the M3 gate.

To reveal possible correlations between MTS modification rates and other parameters including eq. $P_{open}$ (left panel) and probability of being in a cluster, $P_{cluster}$ (right panel), which we assume is a reporter of the M3 gate (see Fig. 2d), we normalized these parameters by the wild-type values (Fig. 5f). MTS modification rates and eq. $P_{open}$ in GluN1 G-A were incongruent, with a reduction in eq. $P_{open}$ despite no change in MTS modification rate (Fig. 5f, left panel). Thus, the divergence in GluN1 G-A eq. $P_{open}$ and MTS modification rate strongly supports the hypothesis that the M2 loop can function as a 'gate'. Furthermore, reductions in $P_{cluster}$ were congruent with MTS modification rates, consistent with the assumption that this parameter is a reporter of the M3 gate (Fig. 5f, right panel). The congruence between $P_{cluster}$ and MTS rate further supports the hypothesis that the M3 gate regulates entry and exit from clusters.

## Discussion

Based on functional experiments and molecular dynamics simulations, we derive two general conclusions about NMDARs (Fig. 6). First, the gating pattern of NMDARs – entry and exit from long- or short-lived closed states – reflects the activity of two gates that directly control the flux of ions through the channel as opposed to modulating this process: a primary 'gate' at the M3 helix bundle crossing that controls entry and exit predominantly from long-lived closed states ($C_4$ & $C_5$),

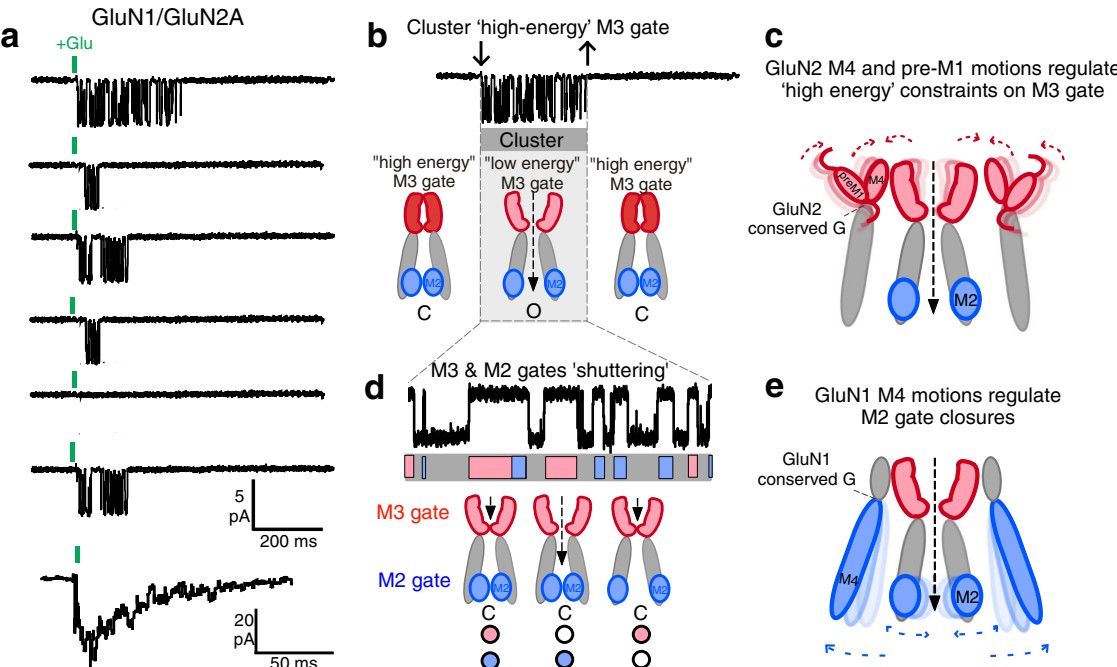

**Fig. 6 | Two gates that are modulated by specific subunits mediate NMDAR synaptic activity. a** Response of NMDARs to brief synaptic like pulses of glutamate. NMDARs enter a 'cluster' of activity. Summed current is shown at the bottom. **b** Entry into and exit from a cluster are driven by a 'high-energy' M3 gate, which presumably opens first for ion channel activity. Structural features driving the high-energy M3 gate include the tension across the M3-S2 linker as well as the positioning of the outer structures – the pre-M1 and M4 helices. **c** GluN2 M4 and pre-M1 regulate entry into and exit from clusters. The status of the M2 loop gate, whether open or closed, can also impact the M3 gate opening (initiation of clusters) as well as their duration. **d** Within a cluster, two gates determine the pattern of activity: a 'low-energy' M3 gate and the M2 gate. We assume that the low-energy M3 gate involves shuttering of just the M3 helices after the pre-M1/M4 triad is already displaced. We assume the M2 gate is the major regulator of the events within a cluster. **e** GluN1 M4, as well as possibly other key structural elements such as the M3 helix, regulates M2 gate including entry into long-lived open states.

and a secondary gate at the M2 pore loop that may regulate mainly short-lived closed states ($C_1$–$C_3$). Our second major conclusion is that these two gates are regulated by the M4 helices of different subunits, with the GluN1 M4 strongly regulating the M2 gate and the GluN2 M4 strongly regulating the M3 gate. These gates are most likely coupled and therefore these actions are not exclusive.

These dual gates lead to a fundamental gating schema. In this schema, entry into and exit from clusters is mediated by an M3 gate that requires a significant amount energy to open, presumably reflecting a required displacement or rearrangement of the GluN2 pre-M1 and M4[35,50–52] (Fig. 6b, c). Notably, the opening of the M3 gate would mediate the initial step in channel activity. Once the M3 helices have rearranged out of a constrained conformation (pre-M1 and M4 displaced), we envision that both the M3 low-energy gate (where outer structures are already displaced) and the M2 gate mediate short-lived closed states (C1, C2, C3) within a cluster (Fig. 6d, e). Although present experiments cannot assign which states are mediated by the M3 or M2 gates, disruption of the M2 gate mostly affects the short-lived states, C1 and C2, while disruption of the M3 gate has the opposite effect, most severely affecting the longest closed states, C4 and C5 (Supplementary Tables 3 and 7).

Our results reveal that the M3 bundle and the M2 loop can function as 'gates', regulating the flux of ions through the pore. Nevertheless, there remain many unresolved issues. While our experiments support the idea that the M3 gate mainly regulates $C_4$ and $C_5$ and the M2 gate mainly regulates the short-lived closed states ($C_1$, $C_2$, and $C_3$), this is by no means absolute since both gates impacts all closed states to some extent. Indeed, it is very likely that the two gates are energetically linked, with the status of the M3 gate, whether opened or closed, impacting the M2 gate and vice-versa. In this scenario, the status of the M2 gate would be one factor that might impact the duration of clusters

(Supplementary Fig. 1d), though this idea needs to be directly tested. In addition, the time course of NMDAR activation is too fast to be solely mediated by the slow C4 and C5 and must involve faster opening events. Whether this reflects a fast component of the activation that is independent of cluster activity is unknown. Finally, the C4 and C5 components are associated with receptor desensitization. Hence, the GluN2 M4 as well as the S2-M4 linker is associated in some fashion with receptor desensitization; however, the mechanisms underlying this association are unknown.

At present we do not know the nature of either gate, whether it is physical occlusion or some other energetic constraint. Given the tight crossing, the M3 gate presumably functions by physical occlusion[8,17]. However, it has been shown that such narrow regions do not necessarily preclude ion flux[3,53]. Although evidence from our MD simulations and macroscopic current recordings suggest that reductions in the pore diameter of the M2 gate correlate with reduced $Ca^{2+}$ permeation[37], this pore loop site is evolutionarily related to the gate in $K^+$ channels[7], which have been shown to occlude ions due to changes in hydrophobicity, or "dewetting"[2]. Thus, it is possible that M3 and M2 gates act via different mechanisms. Further studies are needed to better understand how each gate occludes ion flow.

Subunit-specific regulation (GluN1 vs GluN2) in NMDARs has been identified previously[16,18,43,54]. Here, we showed that the M3 and M2 gates are regulated in a subunit-specific manner, and therefore may be targets for endogenous and pharmacological modulation. For example, zinc's modulatory "rolling" mediated actions on the ATD and LBD that act on the GluN2A M4[55] may act on the M3 gate. Similarly, post translational modifications in the intracellular GluN2 CTD[56,57], which is directly attached to the GluN2 M4, also likely act on the M3 gate. Conversely, modulation of NMDAR activity through the GluN1 CTD, notably the calcium-dependent inhibition of prebound calmodulin[58] is

likely mediated through the M2 gate. Thus, the present work strongly suggests that modulation in NMDARs is achieved in a gate specific manner.

Though it was previously imagined that a single gate generates the distribution of closed and open states seen in MCT and MOT histograms of NMDAR single channels, the introduction of a second gate implies that some of these states are the product of the two gates. Assuming the two gates can enter different structural arrangements partially independent of each other, the number of states observed may be a product of the number of distinct M3 and M2 states. Thus, the presence of two gates may reflect a need for the channel to have more complex control over gating speeds to mediate the wide variety of signaling events that NMDARs control. However challenging isolation of the separate activities of these two gates may prove to channel physiologists, such work is necessary to determine what physiology and pathophysiology each distinct gate mediates. Indeed, pharmacologic tools designed with gate specificity in mind could provide great therapeutic advancements in patients affected by the myriad of NMDAR-associated pathologies[59].

## Methods

### Molecular biology and cell culture

For all experiments, we used rat GluN1 (GluN1a) (NCBI Protein database accession No. P35439) or GluN2A (Q00959) subunits. Numbering includes the signal peptide (GluN1, 18 residues; GluN2A, 19 residues). QuickChange site-directed mutagenesis kits (Agilent) with XL1-Blue super-competent cells were used to generate amino acid substitutions.

Human embryonic kidney 293 (HEK 293) cells (CRL-1573) are obtained from ATCC® approximately every two years. Multiple aliquots of these stocks are made and frozen down in liquid nitrogen. Approximately once per month, we thaw one of these aliquots and use for the subsequent time period. Cells derived from these aliquots display highly consistent morphology and growth patterns. If we have concerns about our stocks of HEK 293 cells, we obtain new stocks from ATCC®.

For details on cell culture and transfection see[34]. Briefly, human embryonic kidney 293 (HEK 293) cells were grown in Dulbecco's modified Eagle's medium (DMEM), supplemented with 10% fetal bovine serum (FBS), for 24 h prior to transfection. Non-tagged cDNA constructs were co-transfected into HEK 293 cells along with a separate pEGFP-Cl vector at a ratio of 4.5:4.5:1 (N1/N2/eGFP) for macroscopic recording, and at a ratio of 4.5:1:1 for single channel recording using X-tremeGene HP (Roche, 06-366). HEK 293 cells were bathed in a media containing the NMDAR competitive antagonist DL-2-amino-5-phosphentoic acid (APV, 100 μM, Tocris) and $Mg^{2+}$ (100 μM). In some instances, we also used 1 mM $Mg^{2+}$. Experiments were performed 18–48 h post-transfection.

### Molecular modeling and MD simulations

The initial structure for the TMD plus LBD-TMD linker construct was built by homology modeling[37]. The sequences of the subunits were from Xenopus laevis GluN1/GluN2B, with two fragments comprising residues L541-P670 (M1-M3 containing) and R794-K841 (M4 containing) for GluN1 and two fragments comprising residues M537-K669 (M1-M3 containing) and G799-Q845 (M4 containing) for GluN2B (all residue numbering based on rat GluN1/GluN2B sequences). The template for homology modeling were the structure of the GluA2 AMPA receptor in the open state (Protein Data Bank entry 5WEO[23]). G-to-A substitutions were introduced at the conserved glycine positions (G815 in GluN1 and G820 in GluN2B) to generate the N1 G-A and N2B G-A constructs.

### Simulations with linkers restrained to an open conformation

The CHARMM-GUI membrane builder server[60] (https://www.charmm-gui.org/) was used to prepare the three constructs in a bilayer with 222 POPC lipids, solvated by 25,201 water molecules (including pore water). All the systems were neutralized using $Na^+$ and $Cl^-$ ions and the final concentration of NaCl was 150 mM. The dimensions of the simulation box were 100.15 Å × 100.15 Å × 127.0 Å. The CHARMM-GUI six-segment protocol was followed to prepare the systems. Specifically, after energy minimization (5000 steps), six short segments of MD simulation were run. The first three segments were 25 ps each with a time step of 1 fs, and the second three segments were 100 ps each with a time step of 2 fs. Segments 1 and 2 were under constant NVT whereas segments 3–6 were under constant NPT. Harmonic restraints were imposed on two dihedral angles and all phosphorus atoms of POPC as well as on the backbone heavy atoms of the proteins. The restraint force constant for the dihedral angles was 250 kcal $mol^{-1}$ $rad^{-2}$ during the energy minimization and segment 1 of simulation but was reduced to 100, 50, 50, and 25 kcal $mol^{-1}$ $rad^{-2}$ in segments 2–5, respectively, and zeroed out in segment 6; the corresponding values for the restraint on the phosphorus atoms were 2.5, 2.5, 1.0, 0.5, and 0.1 kcal $mol^{-1}$ $Å^{-2}$. For the backbone heavy atoms of the proteins, the restraint force constant was 10.0, 5.0, 2.5, 1.0, 0.5, and 0.1 kcal $mol^{-1}$ $Å^{-2}$, respectively, in segments 1–6. Equilibration was performed for 12 ns at constant NPT, in which protein Cα atoms were restrained; the restraint force constant for all the Cα atoms except those on the linker tips was ramped down in the sequence of 5.0, 2.5, 1.0, 0.5, 0.25, and 0.1 kcal $mol^{-1}$ $Å^{-2}$ in 2-ns intervals; the restraint force constant on the linker tip Cα atoms was maintained at 5.0 kcal $mol^{-1}$ $Å^{-2}$. The simulations continued in the production run of 500 ns in 15 replicates for each construct, with only the Cα atoms of the linker tip residues (L541-K544, I667-P670, and R794-E797 in GluN1; M537-R540, L666-K669, and G799-H802 in GluN2B) restrained to the open conformation with a force constant of 5.0 kcal $mol^{-1}$ $Å^{-2}$.

The CHARMM-GUI six-segment preparation was run using Sander in Amber17, and the equilibration and production were run using pmemd.cuda on GPUs[61]. The force field was CHARMM36[62] and the water model was TIP3P[63]. Bonds connected to hydrogens were constrained by the SHAKE algorithm[64]. The cutoff distance for nonbonded interactions was 8 Å; long-range electrostatic interactions were treated by the particle mesh Ewald method[65]. Temperature was kept at 300 K by the Langevin thermostat and pressure was kept at 1 bar by the Berendsen barostat[66].

### Simulations with a transmembrane potential

After the 12 ns equilibration, a second set of 16 replicate simulations was performed under a transmembrane potential of −300 mV[67] for 250 ns at constant NVT. The settings of these simulations were otherwise identical to those without a membrane potential, except that the cutoff distance for nonbonded interactions was 10 Å. Trial simulations were also run under membrane potentials of −100 and −200 mV but were not further analyzed because the ion permeation events were relatively low and no clear distinction from simulations without membrane potential could be drawn.

### Simulations of TMD closure

A third set of MD simulations mimicked the transition of the ion channel from the open state to closed state, by releasing the restraints on the linker tip residues. A snapshot from the first 100 ns of the simulations in the open state without membrane potential was selected for each construct, and simulations in four replicates were restarted without any restraints and run for 3000 ns using pmemd.cuda on GPUs.

### MD analysis

Snapshots were saved in 100 ps intervals, resulting in 5000 and 2500 frames for each open simulation at 0 and −300 mV membrane potential, respectively, and 15,000 frames from the second half of each closure simulation. Pore radii along the pore (or z) axis were calculated

every 10th saved frame using the HOLE (v2.2.005) program[68], and averaged over the replicate simulations. The number of water molecules in the pore was obtained by counting the waters inside the channel from the M3 gate +10 position (V656 in N1 and I655 in N2B) to the M2 top residues (V613 in N1 and L612 in N2B).

Na$^+$ ions that ever visited the pore (i.e., between M3 + 10 and M2 top) were tracked for crossings between three compartments along the permeation pathway. The z coordinate of each such ion was monitored as a function of simulation time. For example, the number of such ions was five in simulation 1 for the wild-type in the open state (z traces shown in Supplementary Fig. 4). A crossing from the vestibule (between M3 + 2 and M2 top) to the extracellular space) was counted only when the ion crossed M3 + 10 and diffused into the extracellular space.

### Single-channel recordings and analysis

Single channel recordings were collected at room temperature (21–24 °C) using an Axopatch 200B (Molecular Devices), filtered at 10 kHz (four-pole Bessel filter), and digitized between 25 and 50 kHz (ITC-16 interfaced with PatchMaster, HEKA). Recording pipettes were pulled from thick-wall borosilicate capillary glass (Sutter Instruments).

On-cell single channel recordings were recorded at steady state. Recording pipettes were fire-polished to final resistances ranging from 10 to 30 MΩs when measured in the bath solution (with an applied positive pipette pressure of ~200 mbar). Seal resistances were between 1 and 15 GΩ. The pipette solution contained (in mM): 150 NaCl, 2.5 KCl, 10 HEPES, 0.05 EDTA, 1 glutamate, and 0.1 glycine, pH 8.0 (NaOH). The high pH and EDTA were used to minimize proton and divalent mediated inhibitory effects, respectively[44]. To elicit inward current amplitudes, we held the electrode voltage at +100 mV. The recordings were ~4–60 min in duration to provide a substantial number of events for analysis. In general, we did not include on-cell patches in analysis unless it contained a minimum of 5000 (typically >50,000) events.

For on-cell single channel analysis, single channel records were idealized in QuB using the segmental k-means (SKM) algorithm with a dead time of 20 μs. Closed and open state fits were performed using the maximum interval likelihood (MIL) algorithm in QuB. Kinetic models of NMDAR gating contain approximately 5 closed states and 3–4 open states[43,44]. For each individual record, state models with increasing closed (3–6) and open (2–4) states were constructed and fitted to the recordings until log-likelihood (LL) values improved by less than 10 LL units/added state or if the next added state showed 0% occupancy[69]. Clusters were identified by a critical time ($T_{crit}$), which was defined by minimizing the false event ratio between the third ($C_3$) and fourth ($C_4$) closed kinetic state[46,70]. To verify single channels in patches, especially for those constructs with a low equilibrium $P_{open}$, we used statistical approaches[71].

Parameters derived include probability of being open ($P_{open}$), mean open (MOT) and mean closed (MCT) times. These measurements were made either for the entire recording period, which is referred to as 'equilibrium' (e.g., equilibrium $P_{open}$), or for just events just during clusters (e.g., cluster $P_{open}$). We also derived the probability of being in a cluster ($P_{cluster}$ = mean cluster length/(mean cluster length + mean intercluster length)).

### Macroscopic current recordings

Macroscopic currents in the whole-cell mode, isolated from HEK 293 cells, were recorded at room temperature using an Axopatch 200B (Molecular Devices), filtered at 2.8 kHz (four-pole Bessel filter), and digitized at 10 kHz (ITC-16 interfaced with PatchMaster, HEKA). Patch microelectrodes were filled with an intracellular solution (in mM): 140 KCl, 10 HEPES, 1 BAPTA, 4 Mg$^{2+}$-ATP, 0.3 Na$^+$-GTP, pH 7.3 (KOH), 297 mOsm (sucrose). The extracellular solution consisted of (mM): 150 NaCl, 2.5 KCl, and 10 HEPES, 0.05 EDTA, pH 8.0 (NaOH). Pipettes had resistances of 2–6 MΩs when filled with the pipette solution and

measured in the standard Na$^+$ external solution. Ca$^{2+}$ was omitted from the extracellular solution to prevent run-down over time. Currents were measured within 15 min of going whole-cell.

External solutions were applied using a piezo-driven double barrel application system. The open tip response (10-90% rise time) of the application system was between 400 and 600 μs. For display, NMDAR currents were digitally refiltered at 500 Hz and resampled at 1 kHz.

### MTS modification reaction rates

To assay the effects of mutations on gating configuration, we calculated the rate of inhibition in agonist-activated macroscopic currents during MTS reagent application. MTS reagents (Toronto Research Chemicals, Toronto, Ontario, Canada) were made as stocks in powder form every day at room temperature and mixed into the recording solution within a few minutes before use. Modification rates were determined at −60 mV by continuous application of MTS reagents in the presence of glycine and glutamate and changes in current amplitudes, which were fitted with a single-exponential function to obtain the time constant (τ). The apparent second-order rate constant for MTS modification rate (k), was related to τ by:

$$k = 1/(\tau[MTS]),$$

where [MTS] is the concentration of the MTS reagent.

### Statistics

Data analysis was performed using IgorPro 7, QuB, and Excel. All average values are presented as mean ± SEM. The number of replicates is indicated in the figure legend or in a table associated with the figure. In instances to determine if outcomes were statistically different from wild-type, we used an unpaired two-tailed Student's t-test to test for significant differences. For multiple comparisons, we used an analysis of variance (ANOVA), followed by the Tukey test. Unless otherwise noted, statistical significance was set at $p < 0.05$. Sample sizes reflects those from prior publications.

### Reporting summary

Further information on research design is available in the Nature Portfolio Reporting Summary linked to this article.

## Data availability

The data that support this study are available from the corresponding authors upon request. Raw datasets generated during the current study as well as beginning and ending pdb of the MD simulations are available at the Open Science Framework repository [10.17605/OSF.IO/Q5B48]. The source data underlying Figs. 2–5 have been provided as a Source Data file, which includes all numbers derived from the analysis of the raw current records and statistics. Previously published PDB files can be accessed under accession codes 6WHR and 5WEO. Rat GluN1a (NCBI Protein database accession No. P35439) and GluN2A (Q00959) were used in this study. Source data are provided with this paper.

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

## Acknowledgements

We thank Donna Schmidt for technical assistance. This work was supported by NIH Grant R01 NS088479 (L.P.W. and H-X.Z.) and an SBU-BNL SEED Grant (LPW).

## Author contributions

J.B.A., M.H., R.P., H-X.Z., and L.P.W. designed research; M.H, J.B.A., and L.P.W. carried out and/or analyzed the functional experiments including single channel recordings, cluster analysis, and MTS applications; R.P., X.L., and H-X.Z., performed the computational studies including modeling and molecular dynamics simulations; J.B.A., M.H., R.P., H-X.Z., and L.P.W. wrote the paper including approving the final version.

## Competing interests

The authors declare no competing interests.
