## [Peer Review File · Nature Communications]

Two gates mediate NMDA receptor activity and are under subunit-specific regulationReviewers' Comments:

Reviewer #1:

Remarks to the Author:

In the manuscript "Two gates mediate NMDA receptor activity and are under subunit-specific regulation" the authors are adding some new pieces into the puzzle of the NMDAR function and regulation.

Although the provided experimental evidence clearly demonstrates the effects of the G-to-A substitutions, the conclusions based on it are often overstated.

The authors suggest a redefinition of the common nomenclature, renaming what is usually known as the selectivity filter to a "(second) gate". In fact it could be even a "third" gate, after the M3 SYTAN and the neighboring "LILI gate". The LILI gate is in principle also consistent with the "GluN1 glycine mainly regulates single channel events within a cluster or bursts of activity, whereas the GluN2A glycine mainly regulates entry and exit from clusters".

To get a better idea about the GluN1 G-to-A-induced changes in the M2 gate/filter pore diameter experimentally, it would be useful to record the single channel properties also in presence of cations with different sizes.

In my opinion, the weakest part of the work is the molecular dynamics and I would suggest either significantly improving the MD setup and analysis, or removing the MD part as it in fact does not provide any additional details to the observed GluN1 G-to-A effects. It is questionable whether a model of the rat GluN1/GluN2B (why not GluN2A?) TMD-only based on an symmetrical AMPA "open" structure represents the GluN1/GluN2A system studied experimentally. The significant rearrangement between and within domains during the open to closed transition in GluN1/GluN2B full length model was studied by MD (10.3390/biom9100546), showing important non-symmetrical switching of N1 and N2B M3 helices induced by extracellular domains.

The receptor is sensitive to the presence of cholesterol in the membrane, using pure POPC could change the TMD behavior. The MD setup is using a rather short electrostatics cutoff of 8Å, possibly lowering the accuracy of simulations especially when interested in ion permeation. There is also no transmembrane potential reported. While the experiments involve Ca²⁺, the MD simulations contain only Na⁺ (at least as described in the methods and "guessing" from descriptions using only "ion" without stating which one). Although the PTMs are somehow mentioned, it was described recently that N2 palmitoylation plays a significant role anchoring the base of M4 helices, however, it was not included in the model.

minor points:

Is there something missing in the "pathway that select ion(s) can cross"?

I don't really know what is meant by "membrane physiology".

In the "'Gates' are barriers that occlude the flux of ions in the closed state.", if the above mentioned closed state model is correct, the gates you are talking about are actually relevant in the open state conformation. But this is probably just a missing definition of the closed state (I guess here you mean liganded (activated) receptor not permeable to ions).

I am not sure the upstream/downstream of transmembrane helices is clearly defined.

In the "A critical residue in the M4 helices", maybe a better would be one of the critical residues.

In the "(eq. Popen) and to the same extent", "and" is not needed or something else is missing?

In the "closed and opens states", just "open"?

I am not very familiar with the current data policy of Nature journals, but the ever more recognized FAIR principles would dictate that the data are deposited in a public repository and not just "available upon request". Although for example the methods section concerning the MD simulations is detailed enough to allow reproducibility, the actual trajectories could be useful to the community.

Reviewer #2:

Remarks to the Author:

The manuscript entitled "Two gates mediate NMDA receptor activity and are under subunit-specific regulation" is a solid study that provides some interesting data that is relevant to the function of the NMDA receptor. It is well written, and summarizes evidence that individual subunits control different aspects of gating, and also provides further support for functional relevance of the M4 in gating. The conclusions that the GluN1 M4 helix controls M2 helix of GluN2 and GluN2 M4 controls GluN1 M3 are important, and simulations suggest that mutations in the M4 region of different subunits can alter the pore diameter at distinct sites. I think it will be of interest to many working on ligand gated ion channels, glutamate receptors, and synaptic transmission. Below are several suggestions the authors should consider that could improve the clarity of the study and perhaps lead to more accurate and useful conclusions.

Major

1. While the data is convincing in showing a change in the long-lived closed dwell time separating clusters of activity, I am not sure the stated conclusion is helpful (from the discussion) "...a primary gate at the M3 bundle helical crossing controls entry and exit from both long lived and short-lived closed states". That means it controls everything—is that the intended conclusion? Also, where are data supporting entry and exit from short-lived states? There are data in supplemental but they are not really discussed.
2. If the 12 sec dwell time reflected an important step in the pathway from agonist binding to channel opening (i.e. gating), then the rise times would be far slower than ~10 ms reported by dozens of authors. However most kinetic studies conclude that these prolonged closed periods in between clusters of activity are some form of agonist-dependent desensitization, and I think a likely conclusion is that the M4 regulates recovery from a desensitized state. Seems like this should be raised or evaluated in terms of double pulse experiments that assess the recovery from the desensitized state.
3. Opening rates have been interpreted in early studies of frog neuromuscular nicotinic receptors from brief "nachs Schlag" closed states. The idea is the pore reverses a gating step during these brief gaps, and then proceeds forward to reopen. If the dwell time reflects a single visitation to one closed state, then the reciprocal of the duration is the rate associated with forward opening rate. Studies from Colquhoun, Auerbach, Traynelis, Gibb and others all showed three brief intra-cluster closed states that were interpreted as a very fast opening (perhaps pore dilation), and two other states with dwell times of ~1 and ~5 ms. If the author wants to conclude something about gating within a cluster, they should discuss some of the changes in various components of the histograms shown in the supplemental material) or provide some modelling results.
4. I think it is important to explicitly state whether the opening of the two gates occurs in any order or in a dependent order (e.g. slow step first, then fast step), as multiple Popescu papers have suggested. Dependent order requires slow step to be first, followed by a fast step and then opening, otherwise there would never be brief sojourns to the closed state in the single channel record if the brief step occurred first and then the slow step immediately preceded opening.

Minor

5. I think the abstracts reference to "clusters" is vague, and bordering on jargon. How many readers interested in synaptic transmission will know what is meant by a cluster? I think the authors need to re-word to a conclusion that a wider audience can grasp and understand.

6. Minor suggested wording change: Page 4 "A variety of disease associated missense mutations have been identified at OR NEAR these conserved glycines...(could choose any of multiple refs to support)

7. Page 4 sentence starting "These data suggest that the M3 gate mediates the longer..." How can you rule out that both gates are important, and because the long dwell time dominates the interval, you are missing changes to the interval with the faster gate changes its rate. That is, both may be required, but you can only see changes in the slow gate (contributing the long delay). A short 10 ms change will hardly alter a 12 second interval.

8. Typo in text on page 6. Duration of long lived state should be sec, not ms in the last sentence.

9. Page 8, first sentence of the second paragraph -- Alasdair Gibb should get the credit for first showing unambiguously that there are five independent closed states in individual activations of a single native NMDA receptors in his clever low concentration experiment (Gibb et al., 1992). The references (40,41) confirmed his conclusions in heterologous systems—Schorge et al I think should be added.

10. There are several papers entirely devoted to the idea that the GluN1 M4 functionally interacts with the GluN2 M3, and the GluN2 M4 interacts with the GluN1 M3 (Chen et al. 2017 exploring actions of a pre-M4 mutation, also Gibb 2018). Chen concludes the interaction of GluN2-M4 with GluN1 M3 is critical for gating, consistent with the conclusions here. Seems as though the conclusions of these papers should be mentioned, which are consistent to some extent with what is being showing here?

11. I would recommend mentioning in the results that saturating agonists are used so a reader doesn't have to turn to the methods or scrutinize the legends.

12. I think it is important to comment that these results were obtained in the absence of divalents. Schorge et al recorded in divalent ions and was unable to reproduce some channel properties observed in the absence of divalents. Alternatively, one could perform a quick experiment in a patch with 0.5-1 mM Ca²⁺ to demonstrate that the conclusions hold in a physiological context.

13. I suspect a subset of readers will want to know what Tcrit was used for cluster analysis when they read about the analysis in the results without stopping their train of thought to turn back to the methods to figure out how you separated clusters, especially for mutations that shortened the long closed duration.

14. Figure 1D: It looks like there are sublevels for the GluN1-G815A mutation, and possibly even the GluN2A-G819A mutation. This seems relevant for a paper discussing pore diameter. Why not show openings at a higher resolution and analyze the sublevels?

15. The K channel literature is fascinating because the authors in the structural era connect their results to work performed before structures were known. Seems like Banke's conclusion ten years ago (2003) before the structure of the NMDA receptor, "These data suggest that NR1 and NR2B subunits, respectively, undergo a fast and slow agonist-dependent conformational change that precedes opening of the pore", is relevant to the conclusions of the current manuscript. I believe that data in Jones et al 2002 with MTSEA also suggests different roles of GluN1 and GluN2 in gating. The current manuscript goes light years beyond these older studies, but it seems generally useful to recognize how early ideas about subunit dependent gating arose in the literature.

16. Near the middle of page 9, could the authors specify that 15 replicate simulations of 500ns each were run per construct? They specify 15 in the Methods section, but also specifying it here would save time from having to go to the Methods and help improve clarity for the reader.

17. In Figure 2 and the related text in Results, positions +2, +6, and +10 aren't defined as being indexed from the S in SYTANLAAF, but it is clarified in the Supplementary info. It would be useful to define it in the text, either by verbally explaining this in the legend for Figure 2 or by moving panel 'a' from Supp Fig 3 to Fig 2 in the main text. I don't see that the magenta sphere is defined in the caption for Fig. 2a. Caption for Fig. 2g refers to left and right panels when the panels are stacked vertically

18. Has the model of the active state used in the MD simulations been used in previous publications? If so, a reference should be given as it would be helpful to see model quality metrics, as well as other information about the simulations such as RMSD plots, included in the supplemental information. If not, perhaps these quality metrics could be added as supplemental information.

19. Given that ion permeation was analyzed in the MD simulations, can the authors comment on why a membrane potential wasn't modeled using a method such as applying a constant electric field or using a polarizable force field? For reference, see <https://doi.org/10.1021/acs.jctc.5b01202>. The simulations as they are model the behavior of ions in the absence of any driving force. If one of these methods for the simulations had been used, I feel that the data presented in Figure 2d and e would be more meaningful and informative. However, it is obviously too much to re-do. Perhaps the authors could add a comment about this in the manuscript as a caveat?

20. Are the values in Figure 2e from a single trajectory, or averaged across the 15 replicates? If they are from a single trajectory, can you explain how that specific trajectory was selected? Can the authors provide an explanation of how individual trajectories were selected for the plots in Supplementary Figure 4?

21. Could the authors add to the Discussion their thoughts on why there are two long-lived closed states if the general claim is that the N2A M4s predominately regulate entry into and exit from long-lived closed states?

22. Page 18: "although evidence from our MD simulations and single channel recordings suggest that reductions in the pore diameter of the M2 gate correlate with reduced ion permeation...", I don't really see evidence of this in the supplementary or main figures at least in terms of single channels for permeation. Maybe I missed something.

23. Y-axes of Fig. 1h and 1i could be more clearly labeled (perhaps 'Closed duration ratio (G-A/wt)'?) Similarly, the word 'closed' should be added between 'mean' and 'durations' in the last sentence on pg. 8. Minor but might help keep the message clear.

Response to Reviewers. Nature Communication

Manuscript#: NCOMMS-22-43338-T. "Two gates mediate NMDA receptor activity and are under subunit-specific regulation"

We greatly appreciate the Reviewers for their thoughtful comments. As outlined below, we have incorporated all comments/suggestions/requests into the revised version of the manuscript in some form. Overall, the comments and suggestions by the Reviewers have greatly improved the readability and clarity of the manuscript.

Please note that we have modified our figure set slightly, splitting original Figure 1 into two figures (new Figure 1 and 2). We did this because we added new information and panels to the original Figure 1, at the request of Reviewers#1 and #2, making it cumbersome as a single figure.

New Figure 1 (structure/cartoon figure) (original Figure 1a-1c). We added new information to the structure/cartoon at the request of Reviewer#1. Making it its own figure we were able to expand its size somewhat making it easier to see all presented information.

New Figure 2 (single channel recordings) (original Figure 1d-1i). We added new panels (Figure 2g-i), as requested by Reviewer#2, and having its own figure accommodated these new panels better.

We present our point-by-point responses to reviews below:

Reviewer #1 (Remarks to the Author):

In the manuscript "Two gates mediate NMDA receptor activity and are under subunit specific regulation" the authors are adding some new pieces into the puzzle of the NMDAR function and regulation.

Although the provided experimental evidence clearly demonstrates the effects of the G to-A substitutions, the conclusions based on it are often overstated.

The authors suggest a redefinition of the common nomenclature, renaming what is usually known as the selectivity filter to a "(second) gate". In fact it could be even a "third" gate, after the M3 SYTAN and the neighboring "LILI gate". The LILI gate is in principle also consistent with the "GluN1 glycine mainly regulates single channel events within a cluster or bursts of activity, whereas the GluN2A glycine mainly regulates entry and exit from clusters".

We apologize if we overstated our conclusions. We are very much aware of the rich history of the structure-function of NMDA receptors, and many aspects of our conclusions are certainly not novel. For example, the idea that the different subunits regulate different aspects of the gating process is widespread in the literature (see also comments of Reviewer#2) and even our earlier experiments indicated this (e.g., Sobolevsky et al., 2007, JGP). Still, we wanted to highlight the idea that the M2 loop can function as a 'gate' – that it can form a barrier for the flux of ions – and that the M3 gate (SYTAN + LILI) and M2 gate are under subunit specific regulation (though we do not believe this is absolute).

In addition, the LILI gate and the SYTAN motif were part of our thought process in trying to conceptualize the nature of the upper M3 gate. We would prefer to continue to lump LILI and SYTAN into the 'primary' gate since even in the original publication the

authors noted that ‘...the LILI motif...form a functional unit with the TTTT ring...’ (Ladislav et al., 2018, FMN).

Nevertheless, we have added a more extensive discussion of the M3 gate – that it is not a single entity – in the Introduction (p. 4) and Discussion (p. 16) and also include the LILI motif in our new structural figure (Figure 1b). We also note that LILI gate is consistent with subunit-specific regulation in the Discussion (p. 16). We appreciate the suggestions.

To get a better idea about the GluN1 G-to-A-induced changes in the M2 gate/filter pore diameter experimentally, it would be useful to record the single channel properties also in presence of cations with different sizes.

The Reviewer brings up an interesting experiment. However, after consideration, we realized that these experiments would be extremely challenging (identifying different conducting states would be extremely hard), and their outcome would be ambiguous in that we are not looking at changes in pore diameter but rather that the M2 functions as a ‘gate’ – it can prevent the flow of permeant ions. The larger sized organics even in the M2 open confirmation would show slow permeation greatly reducing detectability of currents (these sorts of experiments which we have extensive experience with are invariably done in the whole cell mode). In addition, these experiments would more address that the pore dimension is changed, which we previously addressed by looking at changes in Ca²⁺ permeability (e.g., Amin et al., 2018, Nat. Comm.), than whether it acts as a gate.

In addition, these sorts of experiments would also be complicated in interpretation because they would start moving into questions about the nature of channel block. Unfortunately, because the mechanism of pore block is incompletely understood, interpreting the results of such an experiment would be difficult. Our goal in this manuscript was to outline the contribution of the M4s in influencing the activity of two separable gates in the ion channel pore.

In my opinion, the weakest part of the work is the molecular dynamics and I would suggest either significantly improving the MD setup and analysis, or removing the MD part as it in fact does not provide any additional details to the observed GluN1 G-to-A effects. It is questionable whether a model of the rat GluN1/GluN2B (why not GluN2A?) TMD-only based on an symmetrical AMPA "open" structure represents the GluN1/GluN2A system studied experimentally. The significant rearrangement between and within domains during the open to closed transition in GluN1/GluN2B full length model was studied by MD (10.3390/biom9100546), showing important non-symmetrical switching of N1 and N2B M3 helices induced by extracellular domains.

The receptor is sensitive to the presence of cholesterol in the membrane, using pure POPC could change the TMD behavior. The MD setup is using a rather short electrostatics cutoff of 8Å, possibly lowering the accuracy of simulations especially when interested in ion permeation. There is also no transmembrane potential reported. While the experiments involve Ca²⁺, the MD simulations contain only Na⁺ (at least as described in the methods and "guessing" from descriptions using only "ion" without stating which one). Although the PTMs are somehow mentioned, it was described recently that N2 palmitoylation plays a significant role anchoring the base of M4 helices,

however, it was not included in the model.

We appreciate the concerns of the Reviewer. We certainly think the MD simulations are a key part of our conclusions, as we outline below.

First, we acknowledge the significant challenges in realistically simulating the NMDAR open and closed states. As correctly pointed out by the Reviewer, we neglected many details such as cholesterol and PTMs, focusing instead on the features that we felt were the most important for connecting with the electrophysiological studies. Issues about cholesterol and PTM are very important and one we are actively investigating in terms of our model, but these studies go beyond the present study since they add many new experimental manipulations.

Second, we used GluN1/GluN2B because we accumulated a significant number of simulations (some published in the 2018 NC paper but the vast number were unpublished). A large number of simulations is important for making sure the observations we report are robust. Note that our focus is the difference between the wild-type and the G-to-A mutants, and, as we now note (p. 8), the G-to-A mutation in GluN2B produces the same electrophysiological phenotype as in GluN2A (Amin et al., 2018, Nat. Comm.). In addition, we now state in the text (p. 8-9) that the permeant ion is Na⁺, and also note the recordings were done without Ca²⁺ (p. 6, 9).

Third, the open model, based on homology with the AMPAR structure, is not symmetric. The diagonal distance of the GluN2 M3 helices are much longer than the counterpart of the GluN1 M3 helices, in line with electrophysiological data showing greater contribution of the GluN2 subunits to channel gating (e.g., Kazi et al., 2014, Nature Ns). Nevertheless, we did not include the extracellular domains as in Cerny et al., 2019 (new ref. 48). Again, we wanted to carry out as many simulations as possible to increase the number of replicates which was facilitated by using a reduced model.

Fourth, we initially refrained from carrying out simulations under a transmembrane potential because we wanted to isolate the differences between wild-type and the G-to-A mutants without the interference of external factors, especially considering that a large potential (e.g., > 300 mV) might be required to achieve a meaningful number of ion permeation event.

Nevertheless, we agree with the Reviewer (as well as Reviewer 2) and have now made additional simulations with a transmembrane potential. In these simulations we also increased the nonbonded cutoff distance to 10 Å. The outcome of these additional simulations further supports the idea that the M2 loop can form a gate, which is regulated by the GluN1 M4 segment. These new data are included in the Results section (p. 10 in the red-lined version).

minor points:

Is there something missing in the "pathway that select ion(s) can cross"?

We have reworded this sentence.

I don't really know what is meant by "membrane physiology".

We have changed to 'membrane excitability'. Sorry for the ambiguity.

In the "'Gates' are barriers that occlude the flux of ions in the closed state.", if the above mentioned closed state model is correct, the gates you are talking about are actually relevant in the open state conformation. But this is probably just a missing definition of the closed state (I guess here you mean liganded (activated) receptor not permeable to ions).

We have added 'non-conducting' to help clarify 'closed state'. We hope this makes our meaning here clearer.

I am not sure the upstream/downstream of transmembrane helices is clearly defined.

We apologize for the lack of clarity. We now have replaced 'upstream' with 'N-terminal' and 'downstream' with 'C-terminal'.

In the "A critical residue in the M4 helices", maybe a better would be one of the critical residues.

As requested changed to the plural form.

In the "(eq. Popen) and to the same extent", "and" is not needed or something else is missing?

As requested, we have removed the extraneous word 'and' from the text.

In the "closed and opens states", just "open"?

We have corrected the typo. Thanks for catching.

I am not very familiar with the current data policy of Nature journals, but the ever more recognized FAIR principles would dictate that the data are deposited in a public repository and not just "available upon request". Although for example the methods section concerning the MD simulations is detailed enough to allow reproducibility, the actual trajectories could be useful to the community.

We appreciate the Reviewer for pointing out a more rigorous data transparency practice. In our original submission, we included considerable information in the Supplementary Information including extensive tables for many of our figures, figures with additional information as well as additional information on our MD simulations. In the resubmission we have added some more information here. We could post the Excel files associated with these figures/tables but am not sure they would provide anything beyond what is in the figure set and Supplementary Information. We have also followed as far as we know all of the guidelines prescribed by Nature Communications in terms of data transparency.

Reviewer #2 (Remarks to the Author):

The manuscript entitled “Two gates mediate NMDA receptor activity and are under subunit-specific regulation” is a solid study that provides some interesting data that is relevant to the function of the NMDA receptor. It is well written, and summarizes evidence that individual subunits control different aspects of gating, and also provides further support for functional relevance of the M4 in gating. The conclusions that the GluN1 M4 helix controls M2 helix of GluN2 and GluN2 M4 controls GluN1 M3 are important, and simulations suggest that mutations in the M4 region of different subunits can alter the pore diameter at distinct sites. I think it will be of interest to many working on ligand gated ion channels, glutamate receptors, and synaptic transmission. Below are several suggestions the authors should consider that could improve the clarity of the study and perhaps lead to more accurate and useful conclusions.

We appreciate the Reviewer’s positive and insightful comments about the manuscript and have tried to modify the manuscript and its presentation to accommodate the Reviewer’s concern.

Major

1. *While the data is convincing in showing a change in the long-lived closed dwell time separating clusters of activity, I am not sure the stated conclusion is helpful (from the discussion) “...a primary gate at the M3 bundle helical crossing controls entry and exit from both long lived and short-lived closed states”. That means it controls everything—is that the intended conclusion? Also, where are data supporting entry and exit from short-lived states? There are data in supplemental but they are not really discussed.*

We apologize for not making our conclusions and the limitation of our conclusions clearer. Part of the challenge is that there remain many unanswered questions so for certain things we can give ‘conclusions’ and in other instances we can only give ‘best guesses’. Clearly, we did not do a good job of clarifying these issues.

Based on circumstantial evidence (and we continue to work to resolve this issue but it is challenging), we believe that the M2 gate is the major regulator of C1, C2, and C3. At present, our data supports the idea that, at least under our conditions, that the M3 gate mainly regulates C4 and C5 and the M2 gate partially regulates C1, C2, and C3, but that none of these regulatory events are absolute: the M2 gate also impacts C4 and C5 and the M3 gate impacts C1, C2, and C3. Part of the challenge is that the energetics of these gates are most likely linked – the open/closed status of the M3 gate impacts the energetics of the M2 loop and vice-versa. A second challenge is that the M2 gate is ‘behind’ the M3 gate and cannot be studied independently (except for our efforts trying to lock ‘open’ the M3 gate, which has certain limitations). Finally, we believe that the fast energetic events with the initial opening event following initial agonist binding (what determines the activation rate of whole-cell currents) may or may not be structurally

related to the fast closed states measured at steady-state. Again we are trying to test these ideas (using outside-out patches, strategic mutations and various agonists) but these are very extensive experiments and go beyond the present manuscript.

In any case, we have added the data for C1, C2, and C3 to the new Figure 2 (panels g-i) (as well as Supplemental Figure 1) to emphasize that the GluN1 G-A has significant effects on C1, while GluN1 and GluN2A G-A have comparable effects on C2 and C3. We have modified the text, mainly in the Discussion (pp 14-15) to suggest that the M3 gate mainly regulates C4 and C5 while the M2 gate mainly regulates C1, C2, and C3 but that these events are not absolute. We also try to emphasize the complex energetics of these gates. The energetics keeping the M3 gate open (and hence the cluster/burst length) is not only dependent upon the M3-S2/pre-M1/S2-M4, but also the status of the M2 loop and perhaps vice-versa. Hence, the duration of cluster/bursts are also influenced by the energetics of the M2 loop.

2. If the 12 sec dwell time reflected an important step in the pathway from agonist binding to channel opening (i.e. gating), then the rise times would be far slower than ~10 ms reported by dozens of authors. However most kinetic studies conclude that these prolonged closed periods in between clusters of activity are some form of agonist-dependent desensitization, and I think a likely conclusion is that the M4 regulates recovery from a desensitized state. Seems like this should be raised or evaluated in terms of double pulse experiments that assess the recovery from the desensitized state.

We agree completely with the Reviewer – that the S2-M4/M4 is regulating aspects of desensitization, and we apologize for not making this clear in the original submission. We have now expanded on this point in the Discussion (p 15).

Note we did record recovery for desensitization for GluN2A G-A as requested but did not add to the manuscript. We do see an increase in the time required for recovery to occur (1.7 ± 0.2 sec, $n = 8$) for GluN2A G-A versus wild-type (0.97 ± 0.09 sec, $n = 7$). Clearly, the S2-M4/M4 is involved in regulating desensitization but this will require much more detailed investigations to resolve and goes beyond the present study.

3. Opening rates have been interpreted in early studies of frog neuromuscular nicotinic receptors from brief “nachs Schlag” closed states. The idea is the pore reverses a gating step during these brief gaps, and then proceeds forward to reopen. If the dwell time reflects a single visitation to one closed state, then the reciprocal of the duration is the rate associated with forward opening rate. Studies from Colquhoun, Auerbach, Traynelis, Gibb and others all showed three brief intra-cluster closed states that were interpreted as a very fast opening (perhaps pore dilation), and two other states with dwell times of ~1 and ~5 ms. If the author wants to conclude something about gating within a cluster, they should discuss some of the changes in various components of the histograms shown in the supplemental material) or provide some modelling results.

This is an outstanding point and one to be honest we struggle perhaps the most with. As indicated in comments to point 1, we added mean closed duration ratios for C1, C2, and C3 to Figure 2 to better analyze those brief states. We now mention that GluN1 G-A

has a somewhat greater influence on these brief closed states (specially C1) than GluN2A G-A but also note that for C1, C2, and C3 this is not very absolute (in contrast to C4 and C5). Importantly, we wish to emphasize (and have done so in the Discussion) that our interpretation of this data is limited by the high likelihood of energetic coupling between the two gates.

In any case the Reviewer raises some critical considerations that are ultimately essential to resolve to better define the gating mechanism in NMDARs. At present our data set cannot resolve because of numerous complications (see response to Point 1).

4. I think it is important to explicitly state whether the opening of the two gates occurs in any order or in a dependent order (e.g. slow step first, then fast step), as multiple Popescu papers have suggested. Dependent order requires slow step to be first, followed by a fast step and then opening, otherwise there would never be brief sojourns to the closed state in the single channel record if the brief step occurred first and then the slow step immediately preceded opening.

This comment brings up an interesting insight and one we cannot with the present data set directly answer. Nevertheless, we assume that the M3 gate must open first, and we speculate on this in the Discussion, though we note that our data does not directly address, and thus a definitive answer is better left for future studies.

Minor

5. I think the abstracts reference to “clusters” is vague, and bordering on jargon. How many readers interested in synaptic transmission will know what is meant by a cluster? I think the authors need to re-word to a conclusion that a wider audience can grasp and understand.

We thank the Reviewer for this suggestion and agree that many physiologists will not understand the significance of the word. We have added a sentence to the Abstract to better clarify this term. We hope this will make better sense to synaptic physiologists.

6. Minor suggested wording change: Page 4 “A variety of disease associated missense mutations have been identified at OR NEAR these conserved glycines...(could choose any of multiple refs to support)

We thank the Reviewer for suggesting this change, which better acknowledges that mutations near this site may also act via the mechanisms we outline in this work. The change has been made and several citations added (Chen et al., 2017, Mol. Pharm.; Perszyk et al., 2020, JPhysiol).

7. Page 4 sentence starting “These data suggest that the M3 gate mediates the longer....” How can you rule out that both gates are important, and because the long dwell time dominates the interval, you are missing changes to the interval with the faster gate changes its rate. That is, both may be required, but you can only see changes in the slow gate (contributing the long delay). A short 10 ms change will hardly alter a 12

second interval.

As we discuss above and now in the Discussion, we think the gates are strongly energetically coupled. Indeed, C4 and C5 closed durations are significantly changed when we used GluN1 G-A to disrupt M2, just to a much less significant extent compared to GluN2.

8. *Typo in text on page 6. Duration of long lived state should be sec, not ms in the last sentence.*

We thank the Reviewer for catching this error! The text has now been changed.

9. *Page 8, first sentence of the second paragraph -- Alasdair Gibb should get the credit for first showing unambiguously that there are five independent closed states in individual activations of a single native NMDA receptors in his clever low concentration experiment (Gibb et al., 1992). The references (40,41) confirmed his conclusions in heterologous systems—Schorge et al I think should be added.*

We appreciate the Reviewer's insights. As requested, we have added the suggested citations.

10. *There are several papers entirely devoted to the idea that the GluN1 M4 functionally interacts with the GluN2 M3, and the GluN2 M4 interacts with the GluN1 M3 (Chen et al. 2017 exploring actions of a pre-M4 mutation, also Gibb 2018). Chen concludes the interaction of GluN2-M4 with GluN1 M3 is critical for gating, consistent with the conclusions here. Seems as though the conclusions of these papers should be mentioned, which are consistent to some extent with what is being showing here?*

We thank the Reviewer for encouraging a broader discussion of the existing literature. These citations at multiple points have now been added to the Introduction and/or /Discussion.

11. *I would recommend mentioning in the results that saturating agonists are used so a reader doesn't have to turn to the methods or scrutinize the legends.*

We appreciate the suggestion from the Reviewer and have now added this information to the Results section.

12. *I think it is important to comment that these results were obtained in the absence of divalents. Schorge et al recorded in divalent ions and was unable to reproduce some channel properties observed in the absence of divalents. Alternatively, one could perform a quick experiment in a patch with 0.5-1 mM Ca²⁺ to demonstrate that the conclusions hold in a physiological context.*

Both Reviewers aptly point out that our original work could have been clearer about the absence of divalents in our experiments, which was done to give better resolution and to

remove confounding variables from our experiments. Text has now been added (p. 6, 8-9) to the results that directly clarifies this. We also agree that testing Ca^{2+} is a critical question, but feel this is a whole set of new experiments to rigorously address (again which we are trying to do).

13. I suspect a subset of readers will want to know what T_{crit} was used for cluster analysis when they read about the analysis in the results without stopping their train of thought to turn back to the methods to figure out how you separated clusters, especially for mutations that shortened the long closed duration.

We thank the Reviewer for encouraging clarification of our results and to make them more accessible to a wider audience. We now include this information in the Results section (p. 7).

14. Figure 1D: It looks like there are sublevels for the GluN1-G815A mutation, and possibly even the GluN2A-G819A mutation. This seems relevant for a paper discussing pore diameter. Why not show openings at a higher resolution and analyze the sublevels?

The Reviewer points out a possibility we considered previously. There are several problems here. First, the openings for GluN1-G815A are so brief, we are not able to properly and consistently identify any subconductance states. Second, at present it is not clear the nature of subconductance levels in NMDARs and we feel this would require a whole new (and extensive) study to define (which we are trying to do but with mutations that induce more robust subconductance levels).

15. The K channel literature is fascinating because the authors in the structural era connect their results to work performed before structures were known. Seems like Banke's conclusion ten years ago (2003) before the structure of the NMDA receptor, "These data suggest that NR1 and NR2B subunits, respectively, undergo a fast and slow agonist-dependent conformational change that precedes opening of the pore", is relevant to the conclusions of the current manuscript. I believe that data in Jones et al 2002 with MTSEA also suggests different roles of GluN1 and GluN2 in gating. The current manuscript goes light years beyond these older studies, but it seems generally useful to recognize how early ideas about subunit dependent gating arose in the literature.

We apologize for not recognizing these earlier works in terms of the subunit specific effects on gating. We now include them as well as several others in the Discussion.

16. Near the middle of page 9, could the authors specify that 15 replicate simulations of 500ns each were run per construct? They specify 15 in the Methods section, but also specifying it here would save time from having to go to the Methods and help improve clarity for the reader.

As requested, we now include the number of replicates in this section.

17. In Figure 2 and the related text in Results, positions +2, +6, and +10 aren't defined as being indexed from the S in SYTANLAAF, but it is clarified in the Supplementary info. It would be useful to define it in the text, either by verbally explaining this in the legend for Figure 2 or by moving panel 'a' from Supp Fig 3 to Fig 2 in the main text. I don't see that the magenta sphere is defined in the caption for Fig. 2a. Caption for Fig. 2g refers to left and right panels when the panels are stacked vertically

These are very helpful suggestions and appreciate the Reviewer's efforts. We considered moving Supplementary Figure 3a to the main body but realized it would make this figure (Figure 2) even more complicated. We now explicitly state the referencing of +2, +6, and +10 in the text (p. 9). We also now define the magenta sphere in the Figure legend. Finally, the figure legend has been altered to reflect the correct positions of the panels.

18. Has the model of the active state used in the MD simulations been used in previous publications? If so, a reference should be given as it would be helpful to see model quality metrics, as well as other information about the simulations such as RMSD plots, included in the supplemental information. If not, perhaps these quality metrics could be added as supplemental information.

We previously reported the model in Amin et al., 2018, Nature Communications, where many of these metrics, including RMSD plots, were included. This is now noted in the manuscript.

19. Given that ion permeation was analyzed in the MD simulations, can the authors comment on why a membrane potential wasn't modeled using a method such as applying a constant electric field or using a polarizable force field? For reference, see <https://doi.org/10.1021/acs.jctc.5b01202>. The simulations as they are model the behavior of ions in the absence of any driving force. If one of these methods for the simulations had been used, I feel that the data presented in Figure 2d and e would be more meaningful and informative. However, it is obviously too much to re-do. Perhaps the authors could add a comment about this in the manuscript as a caveat?

This is a very valid concern. We initially refrained from carrying out simulations under a transmembrane potential because we wanted to isolate the differences between wild-type and the G-to-A mutants without the interference of external factors, especially considering the fact that a large potential (e.g., > 300 mV) might be required to achieve a meaningful number of ion permeation event. Nevertheless, we agree with the Reviewer (as well as Reviewer 1) and have now made additional simulations with a transmembrane potential. The outcome of these additional simulations further supports the idea that the M2 loop can form a gate, which is regulated by the GluN1 M4 segment. These new data are included in the Results section (p. 10 in the red-lined version).

20. Are the values in Figure 2e from a single trajectory, or averaged across the 15 replicates? If they are from a single trajectory, can you explain how that specific trajectory was selected? Can the authors provide an explanation of how individual

trajectories were selected for the plots in Supplementary Figure 4?

The values in Figure 2e are the total crossing from 15 separate replicates, which now is explained better in the Figure 2 legend.

Trajectories in Figure 2d and Supplementary Figure 4 (they from the same simulation). We selected this example since they all came from the same simulation which showed the most transitions of all the replicates we made. We now explain this in the Supplementary Figure 4 legend.

21. Could the authors add to the Discussion their thoughts on why there are two long-lived closed states if the general claim is that the N2A M4s predominately regulate entry into and exit from long-lived closed states?

This is an outstanding question and one we have thought lots about. Our general idea, and we are certainly not the first to think of this, is that there are multiple structural elements regulating the opening of the M3 gate. The pre-M1 helix, the M3-S2 linkers, the S2-M4/M4 regions, among other structural elements. C4 and C5 presumably reflect to some extent all of these structural elements.

In short we speculate that a modifying process is occurring for our two long lived closed states, with the M4 movement being the primary driver, but with alternative structural confirmations bifurcating this process into two outcomes. However, we cannot delineate the culprit structural elements in this work without further experiments, the exploration of which is for another manuscript.

We have added some of this information to the Discussion (pp 14-15).

22. Page 18: “although evidence from our MD simulations and single channel recordings suggest that reductions in the pore diameter of the M2 gate correlate with reduced ion permeation...”, I don’t really see evidence of this in the supplementary or main figures at least in terms of single channels for permeation. Maybe I missed something.

We have corrected this sentence [I assume what’s meant by this sentence was correlation b/w pore radius and Ca²⁺ permeability as shown in 2018NC].

23. Y-axes of Fig. 1h and 1i could be more clearly labeled (perhaps ‘Closed duration ratio (G-A/wt)’?) Similarly, the word ‘closed’ should be added between ‘mean’ and ‘durations’ in the last sentence on pg. 8. Minor but might help keep the message clear.

We thank the Reviewer for their suggestion and have now altered Figure 1h and 1i to state “Closed duration ratio (G-A/wt)” and changed to ‘mean closed durations’.

Reviewers' Comments:

Reviewer #1:

Remarks to the Author:

Dear Authors, thank you for the revised version of the manuscript. I believe that the introduced changes sufficiently addressed my comments. As already said, the experimental evidence of G-to-A effects is solid and I understand that performing complicated new experiments or modified MD simulations at time scales comparable to the GluN2B ones would be impractical now.

You are right that the Supplementary material is rich and the data is treated according to the rules. My appeal was more like "it would be nice to have the raw trajectories as well". It would, like for example in structural biology field, allow reanalysis or even more importantly development or benchmarking of new data processing methods. Obviously the infrastructure allowing efficient storage of large datasets in an interoperable way is not very mature (neither zenodo nor say model archive are perfect for this purpose yet) but we are in a hen and egg situation here, without researchers offering their raw data, the repositories will evolve only slowly (if ever) into a useful and user-friendly data resource.

Reviewer #2:

Remarks to the Author:

The authors revised manuscript has adequately addressed the points raised. Congratulations on the completion of a very nice study.